# Development of Multifunctional Detection Robot for Roller Coaster Track

**DOI:** 10.3390/s23208346

**Published:** 2023-10-10

**Authors:** Weike Song, Zhao Zhao, Kun Zhang, Huajie Wang, Yifeng Sun

**Affiliations:** 1Key Laboratory of Special Equipment Safety and Energy-Saving for State Market Regulation, China Special Equipment Inspection and Research Institute, Beijing 100029, China; songweike@csei.org.cn (W.S.); zhangkun@csei.org.cn (K.Z.); wanghuajie@csei.org.cn (H.W.); sunyifeng@csei.org.cn (Y.S.); 2China Special Equipment Inspection and Research Institute, Beijing 100029, China

**Keywords:** multifunctional detection robot, track defect detection, vision inspection, electromagnetic ultrasonic, eddy current

## Abstract

Recent advances in roller coasters accelerate the creation of complex tracks to provide stimulation and excitement for humans. As the main load-bearing component, tracks are prone to damage such as loose connecting bolts, paint peeling, corroded sleeper welds, corroded butt welds, reduced track wall thickness and surface cracks under complex environments and long-term alternating loads. However, inspection of the roller coaster tracks, especially the high-altitude rolling tracks, is a crucial problem that traditional manual detection methods have difficulty solving. In addition, traditional inspection is labor-intensive, time-consuming, and provides only discrete information. Here, a concept of the multifunctional detection robot with a mechanical structure, electrical control system, camera, electromagnetic ultrasonic probes and an array of eddy current probes for detecting large roller coaster tracks is reported. By optimizing the design layout, integrating multiple systems and completing machine testing, the multifunctional roller coaster track detection robot exhibits outstanding performance in track appearance, thickness and crack detection. This study provides great potential for intelligent detection in amusement equipment, railcar, train and so on.

## 1. Introduction

Amusement facilities including the roller coaster, Ferris wheel, pendulum ride, etc., are all manned devices, which bring excitement and stimulation to tourists. However, once an accident occurs, it can cause significant casualties and extremely adverse social impacts. According to accident events from 38 countries, roller coasters account for nearly half of accidents, and the most common event type was malfunction [1]. As the main load-bearing component of a large roller coaster under harsh outdoor environments and the alternating load, the track has potential safety hazards, such as loosening of connecting bolts, peeling of paint, cracking of sleeper welds, cracking of track butt welds, wear of track pipes, and corrosion thinning. Furthermore, the track of the roller coaster is difficult to manually detect because of its high and complex trajectory.

To resolve this problem, detection robots have attracted great attention in many fields. In the welding field, multiple sensors integrated with a wall-climbing robot have been developed for magnetic particle testing, which achieves collaborative control of the highly integrated robot and efficient detection [2]. Robots for circumferential crack-like defect detection of pipelines were investigated, which makes it easy to inspect irregularly shaped welds [3]. Additionally, some studies have integrated detection sensors and cleaning equipment into robots and developed a series of ship detection and cleaning robots [4,5,6,7,8]. In the vision-based detection field, a vision-detection robot for narrow butt joints has been proposed, and the 3D position of the joint can be calculated [9,10]. As for measurements of the seam gap, the real-time image acquisition robot was developed for the detection of welding seam location and characteristics [11]. Li et al. proposed a robust automatic welding seam identification and tracking method by utilizing structured-light vision, where the string obtained from the object image matches the string in the model to determine the position of the weld seam [12]. A passive visual sensing method and an improved Canny operator for calculating the deviation of joints relative to the welding torch have been studied by Xu et al. [13].

Current research on track detection mainly focused on a single system, such as detecting track faults and monitoring [14]. Additionally, the detection is only applicable to flat railways without rolling. For example, a noncontact inspection system was developed for noncontact rail integrity evaluation, which is based on air-coupled guided wave probing [15]. Acoustic Emission (AE) is also used for monitoring and investigating the growth of cracks in railway lines [16]. The application of visual detection technology based on image processing in the railway industry has been summarized by Liu [17]. Based on the measurement values of the wavelet spectrum analysis algorithm, the vertical acceleration of the train can be used to detect rail sinking [18]. Some researchers have developed railway inspection robots for the detection of railway tracks, where they have conducted experiments in many studies by arranging various types of fault detection sensors and detection equipment [19,20]. For example, the Canadian National Railway Company (CN) has developed an automatic rail car that can wirelessly measure the geometric shape of the track [21]. Ultrasonic testing can be performed manually or be integrated into the testing system on the train for detection tracks using image recognition and object detection methods [22,23]. According to the track category, the UK railway network adopts this inspection method. An eddy current installed on the train has been developed as an effective method for detecting cracks on railway tracks [24]; it can detect the inhomogeneity that exists in the metallic surface. The electromagnetic acoustic transducer system detection for detecting surface and internal defects of rails has been reported in recent years [25,26,27], which generate eddy currents on the surface layer of the workpiece with the same frequency but opposite directions based on Faraday’s law of electromagnetic induction.

So far, there is no multi-functional robot that can integrate visual inspection and non-destructive testing functions for detecting rolling tracks. Here, a multifunctional climbing detection robot with a mechanical structure, electrical control system, wireless control and transmission module, camera, electromagnetic ultrasonic probes and array eddy current probes was designed. Eventually, visual inspection of the appearance, wall thickness inspection of the track tube and surface crack detection were carried out on a certain vertical roller coaster track; the detection robot can stably move along the rolling track and detection results were basically consistent with manual retesting results, which verified the effectiveness and accuracy of the multi-functional detection robot developed in this paper. This study also provides a reference for future multi-functional intelligent detection of the large roller coaster track. In Section 2, the mechanical and the electrical control system of the multifunctional roller coaster track detection robot with different sensors are established and tested. Finally, the conclusions are drawn in Section 3.

## 2. The Architecture of the Multifunctional Roller Coaster Track Detection Robot

In recent years, detection robots with various sensors [14] have become increasingly popular, including visual sensors, light detection, ultrasonic sensors, thermal sensors, laser vision systems, eddy currents, electrical acoustic transducers, etc. The advantage of these detection robots is to rescue human inspectors from dangerous environments and tedious inspection work. Meanwhile, these robots provide more standardized and objective inspections than human inspectors and reduce inspection costs. Based on these studies, the multifunctional detection robot for the roller coaster track has been developed in this paper for the first time.

Roller coaster tracks are located outdoors and subjected to complex long-term alternating load conditions. Based on the characteristics and analysis of failure modes of these tracks, the multifunctional track detection robot developed in this paper mainly includes mechanical, electrical control, video detection, electromagnetic ultrasonic thickness detection and array eddy current crack detection subsystems. Therefore, some key detection items of the track can be measured, such as loose connection bolts, paint peeling, corroded sleeper welds, corroded track butt welds, reduced track wall thickness and surface cracks.

The multi-functional inspection robot adopts a roller coaster wheel system structure, an additional drive system, a clamping system, an inspection system, a control system, etc., as illustrated in Figure 1. The wheel system structure is the main load-bearing component, and corresponding systems are installed on it to meet the requirements of autonomous walking and detection. It mainly consists of three parts: the body structure, electrical control system, and detection system. Table 1 shows the basic parameters and functional requirements.

The multi-functional detection robot developed for roller coaster tracks is applied to a vertical roller coaster to detect the appearance, wall thickness, and surface cracks of the tracks. The vertical roller coaster track has various types such as spiral rings, vertical rings, and horizontal rings, as shown in Figure 2a, where the detection situation is shown in Figure 2b.

### 2.1. Mechanical Structure

The vehicle structure consists of seven main parts: frame, front axle assembly, rear axle assembly, front wheel set, rear wheelset, clamping mechanism and driving system, as shown in Figure 3.

The main contribution that is not offered by equivalent state-of-the-art systems is the dual-servo motor-controlled walking system. The roller coaster track is usually two sets in parallel, where the left and right driving wheels of the vehicle walk along the corresponding side tracks. However, the traditional motor-driven control method applied to this type of rail vehicle has the following problems: whether the left and right drive wheels are driven by the same motor through a differential transmission mechanism, or are independently driven by two motors, there is a problem of two wheels interacting with each other. Therefore, the relative motion state between the left and right wheels and the track is inconsistent, ultimately leading to unstable or even abnormal vehicle walking, which affects the progress of inspecting work.

To solve the problem of asynchronous walking, servo motors 1 and 2 are used to control the rotation of drive wheels 1 and 2, respectively. Meanwhile, the rotation of servo motor 1 is controlled and the current feedback value of servo motor 1 is obtained, then the torque of servo motor 2 is adjusted based on the current feedback value of servo motor 1. Additionally, the current feedback value of servo motor 2 needs to be obtained in real time, and the difference between the current feedback values of servo motors 1 and 2 should be calculated, and then the output torque of servo motor 2 should be controlled based on this difference. The control objective is to ensure that the current feedback values of servo motor 1 and servo motor 2 are equal. Control the output torque of servo motor 2 using PID algorithm. Set the difference e between the current feedback values of servo motor 1 and 2 to be I_1_−I_2_, where I_1_ and I_2_ are the current feedback values of servo motor 1 and 2. When e > 0, the proportion coefficient K_p_ in the PID algorithm is adjusted according to the size of e. For example, the smaller the value of e, the smaller is K_p_ to reduce the static error of the system and weaken fluctuations; when e < 0, the static error of the system is reduced through the integration in the PID algorithm.

The vehicle body adopts a typical full-degree-of-freedom roller coaster chassis structure with the front axle connected to the frame through joint bearings, and the rear axle fixed to the frame through a vertical axis, ensuring that the detection device has sufficient degrees of freedom. The driving system adopts a DC servo motor, which is connected to the reducer through a synchronous belt to drive the main walking wheel to move. The clamping mechanism adopts a cylinder and connecting rod mechanism, which can provide sufficient and reliable compression to the driving wheel. It ensures that the driving wheel is always effectively pressed on the track surface and maintains stable compression force under different sporty postures. In order to verify whether the mechanical structure can adapt to the track characteristics, the main test items including the structure operation interference test and the track-climbing achievement test were carried out, which can successfully achieve various operational functions.

### 2.2. Electrical Control System

The electrical control system, including the onboard control system (Figure 4a) and the handheld operation terminal (Figure 4b), is used to control the operation of the track detection device and vehicle. Based on the handheld operating terminal and wireless communication, remote control operations such as walking, stopping, advancing, retreating, and speed regulation of the track detection device can be achieved. Additionally, wireless communication can remotely control the start and stop of video visual inspection devices, electromagnetic ultrasonic thickness detection devices, and array eddy current crack detection devices. The dual display screen of the handheld operating terminal can display a real-time display of thickness detection data, crack detection data, vehicle status information, and position information, etc. The onboard control system memory and handheld operation terminal memory achieve real-time storage of video, detection data, vehicle status, track position and other information, as well as real-time alarm and event recording of exceeding standard data. Meanwhile, the battery can provide a stable power supply for the whole structure. See Table 2.

### 2.3. Visual Inspection System

The macroscopic information on the track surface, such as loose connecting bolts, paint peeling, corroded sleeper welds, and corroded track butt welds, can be detected by a video visual inspection system. The robot designed in this paper mainly includes the front-end Gopro camera, local camera, pan tilt servo motor, video-processing module, etc., as shown in Figure 5a. 

The GoPro camera at the front of the vehicle (Figure 5b) collects macroscopic information about the track environment, which can be adjusted through two pan-tilt servo motors for the left, right, and pitch angles. Four local cameras (Figure 5c) are set up on the contact surfaces of the walking wheels and the side wheels of the two tracks to collect local detailed information on the track surface. Macro and local video information (Figure 5d,e) is transmitted to the video-processing module, while the onboard controller transmits real-time location information to the video processing module. The video processing module synthesizes video information and location information to form video information based on location information. The onboard controller transmits video information to the handheld operating terminal controller through a wireless image transmission module. 

Through visual inspection of the entire video, it can be seen that the macroscopic state of the roller coaster track is in good condition, with no obvious damage or failure, as shown in Figure 5d. However, Figure 5e shows that the wear of the track is relatively severe near the 90° position of the vertical ring via a local video image.

### 2.4. Electromagnetic Ultrasonic Thickness Detection System

The electromagnetic ultrasonic thickness measurement method used in this paper is the pulse reflection method. The probe generates ultrasonic waves that propagate perpendicular to the surface. Ultrasonic waves will reflect when they reach the lower surface of the test track wall, which continue to propagate upwards along the thickness direction and are then received by the receiving probe on the upper surface.

The electromagnetic ultrasonic thickness-detection system, including electromagnetic ultrasonic thickness-measurement probes, data acquisition cards, thickness-measurement processing units and so on, is used for detecting the wall thickness of track pipes, as shown in Figure 6. Electromagnetic ultrasonic thickness measurement probes 1 (left rail) and 2 (right rail) transmit ultrasonic pulses of track pipe wall thickness to data acquisition cards 1 and 2, respectively. Data acquisition cards 1 and 2 convert ultrasonic pulse signals into voltage signals and transmit them to the thickness measurement processing module, where the voltage signals are calculated, analyzed, and processed to generate thickness data for both tracks. Meanwhile, the position module obtains track position information and transmits it to the onboard controller, which is then transmitted to the thickness-measurement processing unit. The next step is determining the thickness data based on position information sent to the onboard controller, then the handheld operating terminal controller receives it through a wireless control module, which is displayed in real time on the display screen of the human–computer interaction interface and stored in the memory of the handheld operating terminal, facilitating offline data processing and analysis.

The electromagnetic ultrasonic thickness detection system adopts a dual-channel set, mainly consisting of two electromagnetic ultrasonic probes, probe-clamping mechanisms, and data-processing modules. The clamping mechanism fixes the electromagnetic ultrasonic probe between the two active wheels of the front wheel group of the detection device. In order to adapt to various transformation types of the track, the clamping mechanism has two degrees of freedom in both directions to prevent interference between the probe and the track, where probes are stably pressed onto the surface of the track to ensure a certain lift off the gap, thereby maintaining the stability of the probe operation. The fixed type and detection data display of the electromagnetic ultrasound probe are shown in Figure 6b,c. To enhance the accuracy of detection, the time difference Δ*t* between *n* echo signals can be used to calculate the thickness *d*, which is given by
(1)$$d=\frac{\Delta t\cdot v}{2\left(n-1\right)}$$

The thickness detection results of the track are shown in Figure 6d, it can be seen that thickness exceeds the standard value (10 mm) within the ±5 mm range at certain positions of the left and right tracks. Due to the presence of foreign objects on the track surface during the actual operation of the detection device, some data may have inaccuracies. The track thickness at the location is manually retested where there was a deviation in the track thickness. Generally, positive deviation data can be removed, and the focus is on thickness retesting at the location of thickness reduction.

### 2.5. Array Eddy Current Crack Detection System

The principle of eddy current detection in this paper is that when alternating current is applied to the eddy current detection coil, an alternating magnetic field can be generated. It can induce eddy currents on the surface of the roller coaster track, which generates a magnetic field that interacts with the original coil magnetic field. When there are defects on or near the track surface, the change in induced eddy current will lead to the change in impedance or voltage of the detection coil, which can achieve defect detection on the track surface and near surface. Therefore, the array eddy current crack detection system includes the array eddy current detection module, data-acquisition card, crack-processing unit, etc., as shown in Figure 7a.

Array eddy current detection modules 1 (on the left rail), 2 (on the left rail side), 3 (on the right rail side) and 4 (on the right rail side) transmit the eddy current detection signal on the track surface to the data-acquisition cards, which convert the eddy current detection signal into a voltage signal and transmit it to the crack-processing module for calculating, analyzing and processing. Then, the information on the crack of the track is generated with track position information from the position module and transmitted to the onboard controller, where crack information based on the position coordinates from the crack-processing unit is transmitted to the handheld operating terminal controller through a wireless control module. Finally, it is displayed in real time on the screen of the human–computer interaction interface and stored in the memory of the handheld operating terminal, which is beneficial for offline data processing and analysis.

The array eddy current crack detection system detects cracks on the main walking surface and side wheel walking surface of the track, using a 16-channel array eddy current probe. According to the principle of eddy current, the induced eddy current density in the tested conductor is
(2)$$B\left(z\right)=B_{0}\left(z\right)\exp\left(-z/\sqrt{\frac{1}{\pi f\mu \sigma }}\right)$$
where *B* and *B*_0_ are the surface current density and the eddy current density. *z* is the distance from the surface of the conductor. *f* is the frequency of the eddy current excitation signal. *μ* and *σ* are the magnetic permeability and conductivity of the tested conductor, respectively. The eddy current penetration depth *δ* can be obtained by
(3)$$\delta =\sqrt{\frac{1}{2\pi f\mu \sigma }}$$

Therefore, the parameters of the probe are determined based on the required crack depth to be detected, where a stable fixed height is crucial. Each probe integrates four channels and is fixed to the front wheel group of the detection device through a probe clamping mechanism. Four sets of pressure springs and pressure wheels are used to ensure stable contact and lifting height between the probe and the track surface. The array eddy current probe and its fixing type are shown in Figure 7b,c. The array eddy current crack-detection system scans the crack information on the track surface in real-time, and the data display interface is shown in Figure 7d.

The surface crack detection results of the track have a large amount of data, and this paper only displays the location signal of the discovered defect, as shown in Figure 7d. It can be observed that a defect was found at 30 m of the track from channels S7, S14, S15 and S16. But, the specific morphology of the defect is uncertain, which requires further testing at this location. Through on-site polishing and magnetic particle testing, it was found that there were uneven pit defects rather than obvious crack defects, as shown in Figure 7e.

## 3. Conclusions

In summary, a multifunctional inspection robot suitable for roller coaster tracks is reported. Firstly, a vehicle structure was developed that can actively drive and controllably walk on roller coaster tracks. On this basis, a vehicle electrical control system with remote wireless drive control, detection control, data wireless transmission, display, storage and other functions was developed and tested. In terms of testing functions, a video visual inspection system with macro and local video detection of the track has been integrated and tested. Additionally, an electromagnetic ultrasonic thickness-detection system and array eddy current crack-detection system have been equipped, which achieve the wall thickness of the track and surface-crack-detection functions. Finally, the intelligent robot with multiple detection functions visualized the appearance, wall thickness detection, and surface crack detection of a vertical roller coaster track, which were basically consistent with manual retesting results, verifying the effectiveness and accuracy of the multi-functional detection robot for large roller coaster tracks. It should be noted that due to the lack of automatic fault identification, the current version of the detection robot relies on manual assistance. As a result, the potentiality of the multifunctional detection robot as a new detection method targeting tracks with complex trajectories has been exhibited.

## Figures and Tables

**Figure 1 sensors-23-08346-f001:**
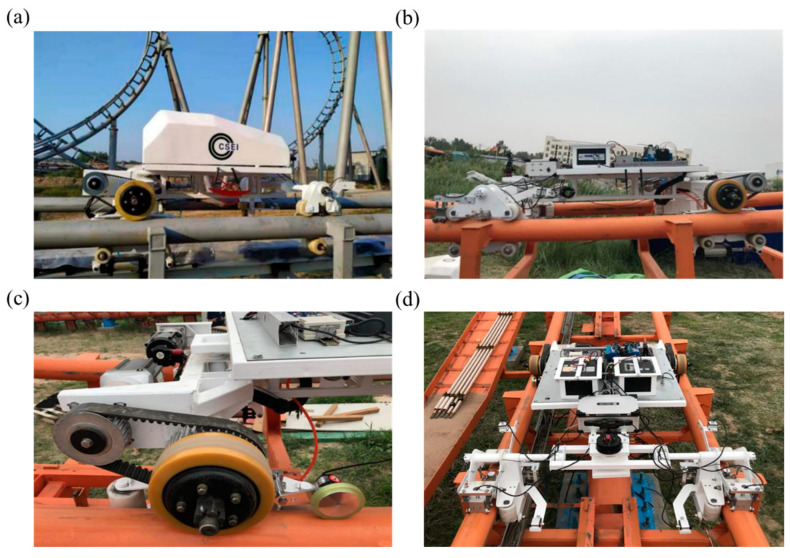
(**a**) Appearance, (**b**) side layout, (**c**) wheel structure and (**d**) top view layout of the multifunctional inspection robot for a roller coaster track.

**Figure 2 sensors-23-08346-f002:**
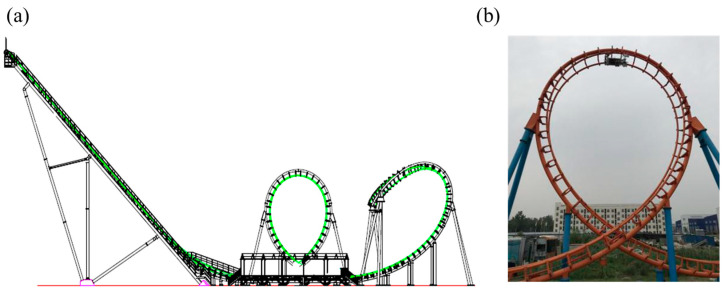
(**a**) Structure diagram and (**b**) multifunctional inspection robot on a certain vertical roller coaster track.

**Figure 3 sensors-23-08346-f003:**
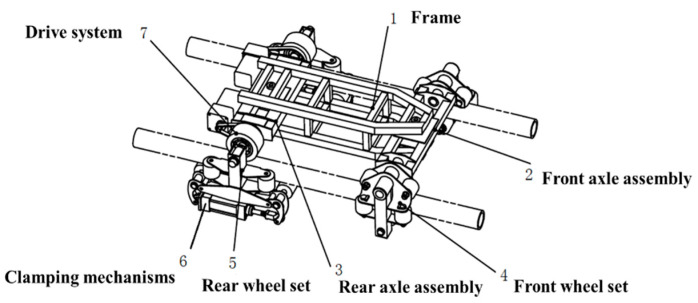
Structure diagram of the vehicle structure.

**Figure 4 sensors-23-08346-f004:**
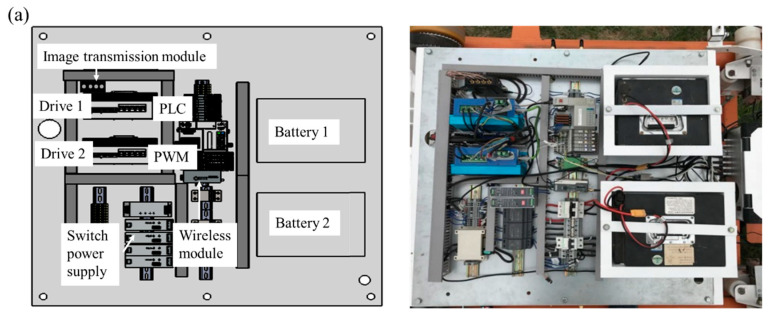
(**a**) The onboard control system and (**b**) the handheld operation terminal.

**Figure 5 sensors-23-08346-f005:**
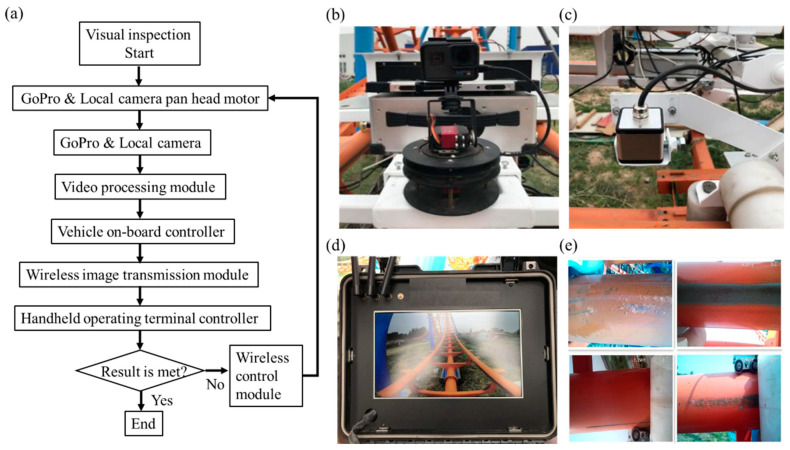
(**a**) Structural framework diagram of video visual inspection system; (**b**) macro and (**c**) local video camera layout; (**d**) macro and (**e**) local video image.

**Figure 6 sensors-23-08346-f006:**
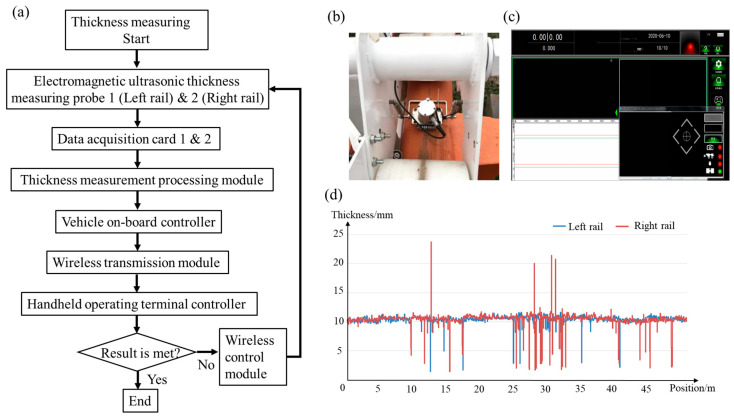
(**a**) Structural framework diagram of the electromagnetic ultrasonic thickness measurement system; (**b**) fixed types of electromagnetic ultrasound probes; (**c**) detection data display and (**d**) track thickness test results.

**Figure 7 sensors-23-08346-f007:**
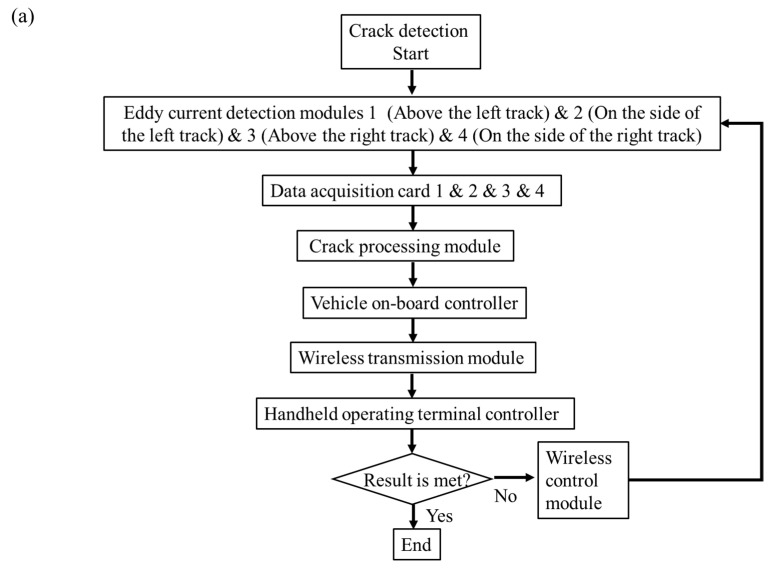
(**a**) Structural framework diagram of the array eddy current crack detection system; (**b**) array eddy current probe and (**c**) fixed type; (**d**) inspection results of track surface crack; (**e**) pit condition of track surface.

**Table 1 sensors-23-08346-t001:** Technical parameters and indicators of the multifunctional inspection robot for roller coaster track.

Item	Technical Index
Applicable environment	−10~40 °C
Running speed	0.3 m/s, 0.2 m/s, 0.1 m/s
Climbing ability	0~360°
Maximum bearing capacity	350 kg
Applicable track types	Rail spacing: 700~1220 mm Rail tube diameter: 89~140 mm
Detection function	Macro and detail video detection of tracks, track thickness detection and surface crack detection

**Table 2 sensors-23-08346-t002:** The hardware description, model, parameter and brand of the control and drive systems used.

Location	Name	Model	Parameter	Brand
Onboard control system	Power supply battery	6-EVF-32	12 V 32 Ah	Tian Neng
	Battery Charger	Local	Suitable for 48 V 32 Ah batteries	Chao Wei
	PLC	1769-L16ER-BB1B	Dual Ethernet w/DLR capability, 6 I/O Expansion via 1734 POINT I/O, 24VDC	AB
	I/O	1734 series	1*8-point switching input and output, 1*4-point analog input and output	AB
	Servo Driver	DCPC-09050-E	DCPC-09050-E	DMK
	Servo motor	130M-09525C5-EB	2 KW, 48 V, 2500 RPM, 6 NM	DMK
	Gear reducer	PLX142-L3-120-S2-P2	120:1 planetary reducer	DMK
	Wireless station	WLAN5100	IEEE 802.11 a/b/g/n, WLAN access point, supporting 2.4 G and 5 G, 300 Mbps	Phoenix
	Camera	GoPro	High definition 4 K motion camera	GoPro
	Wireless image transmission	Local	500~700 M transmission	Freecast
Handheld operator	Power supply battery	Local	24 V 8 Ah	Local
	Battery Charger	Local	Suitable for 24 V 8 Ah batteries	Local
	Touch computer	PPC-3100S	10 inch touch screen, dual Ethernet card, Win 7	Adventach
	Wireless station	WLAN1100	IEEE 802.11 a/b/g/n, WLAN access point, supporting 2.4 G and 5 G, 300 Mbps	Phoenix
	Wireless image transmission	Freecast CP3039	500~700 M transmission	Freecast

## Data Availability

The datasets used in this study are all public datasets.

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
