# Peer review of "Development of Multifunctional Detection Robot for Roller Coaster Track"

_sensors, 2023, doi:10.3390/s23208346_

Round 1

Reviewer 1 Report

The reviewed work is an interesting example of solving the problem of automating the process of detecting and identifying structure damage. From the scientific point of view, the work does not bring any elements of novelty because the  idea of inspection robots is well known, and the methods used: visual inspection, eddy current method and ultrasonic method are well known and used for years in diagnostic systems.

The undoubted advantage of the work is the practical combination of these methods and their integration with the working platform that allows to automate the inspection process of a complex object such as a roller coaster track.

When it comes to detailed comments, it is necessary to correct the drawings so that they become more readable. They are too small in their current form.

To sum up, if the journal's policy is to publish engineering/practical works, then the work is most suitable for publication. However, if scientific aspects are also important, it is difficult for me to find elements of novelty that would allow me to publish my work.

My initial decision is positive, as long as the journal's policy allows for the publication of works of a constructional and practical nature. Nevertheless I leave the final decision to the editors.

Author Response

When it comes to detailed comments, it is necessary to correct the drawings so that they become more readable. They are too small in their current form.

Our replies:

We thank the reviewer’s constructive comment. We have adjusted the drawings layout in the revised manuscript.

Reviewer 2 Report

The paper describes the development of a vehicle of inspection for roller coaster track. It is a relevant topic, but we are facing an application project, without major scientific novelties. What are the main contributions that are not offered by equivalent state-of-the-art systems?

It is necessary to improve the state of the art with reference to works in line with what is described in this work. The second chapter should be the review of the state of the art. The paper has few references and some of them are self-citations by the authors and this should also be avoided.

All flowcharts need to be revised. Flowcharts have a beginning and an end, and all steps have a logical, interconnecting sequence. What it presents are just boxes of functional modules without realizing what makes it pass from one stage to another.

The hardware description needs to be made, with clear evidence of the controller used, characteristics and processes carried out.

Inspection is limited to viewing an image from a GoPro or an analysis from an ultrasound system. Where is the automatic fault identification?

minor issues:

- Avoid writing in the first person - "We";

- The structure of the paper should be written at the end of the introduction;

- Figure 1, identification of each subfigure with a), b), c) and d);

Author Response

Reviewer #2:

The paper describes the development of a vehicle of inspection for roller coaster track. It is a relevant topic, but we are facing an application project, without major scientific novelties.

  1. What are the main contributions that are not offered by equivalent state-of-the-art systems?

Our replies:

We thank the reviewer’s constructive comment. The main contribution that is not offered by equivalent state-of-the-art systems is the dual servo motor-controlled walking system. The roller coaster track is usually two sets in parallel, where the left and right driving wheels of the vehicle walk along the corresponding side tracks. However, the traditional motor-driven control method applied to this type of rail vehicle has the following problems: whether the left and right drive wheels are driven by the same motor through a differential transmission mechanism, or are independently driven by two motors, there is a problem of two wheels interacting with each other. Therefore, the relative motion state between the left and right wheels and the track is inconsistent, ultimately leading to unstable or even abnormal vehicle walking, which affects the progress of inspecting work.

Our revisions:

In Section 2.1, we added:

To solve the problem of asynchronous walking, servo motors 1 and 2 are used to control the rotation of drive wheels 1 and 2, respectively. Meanwhile, the rotation of servo motor 1 is controlled and the current feedback value of servo motor 1 is obtained, then the torque of servo motor 2 is adjusted based on the current feedback value of servo motor 1. Additionally, the current feedback value of servo motor 2 needs to be obtained in real-time, and the difference between the current feedback values of servo motors 1 and 2 should be calculated, and then the output torque of servo motor 2 should be controlled based on this difference. The control objective is to ensure that the current feedback values of servo motor 1 and servo motor 2 are equal. Control the output torque of servo motor 2 using PID algorithm. Set the difference e between the current feedback values of servo motor 1 and 2 to be I1-I2, where I1 and I2 are the current feedback values of servo motor 1 and 2. When e>0, the proportion coefficient Kp in the PID algorithm is adjusted according to the size of e. For example, the smaller the value of e, the smaller Kp to reduce the static error of the system and weaken fluctuations; When e<0, the static error of the system is reduced through the integration in the PID algorithm.

  1. It is necessary to improve the state of the art with reference to works in line with what is described in this work. The second chapter should be the review of the state of the art. The paper has few references and some of them are self-citations by the authors and this should also be avoided.

Our replies:

We thank the reviewer’s constructive comment. We have improved the state of the art with reference with revisions highlighted in red in the Introduction. We have added 10 references on track detection in the revised manuscript.

Our revisions:

In Introduction, we added:

Some researchers have developed railway inspection robots for the detection of railway tracks, where they have conducted experiments in many studies by arranging various types of fault detection sensors and detection equipment [19, 20]. For example, the Canadian National Railway Company (CN) has developed an automatic rail car that can wirelessly measure the geometric shape of the track [21]. Ultrasonic testing can be performed manually or integrated into the testing system on the train for detection tracks using image recognition and object detection methods [22, 23]. According to the track category, the UK railway network adopts this inspection method. Eddy current installed on the train has been developed as an effective method for detecting cracks on railway tracks [24], it can detect the inhomogeneity that exists in the metallic surface. The electromagnetic acoustic transducers system detection for detecting surface and internal defects of rails has been reported in recent year [25-27], which generate eddy currents on the surface layer of the workpiece with the same frequency but opposite directions based on Faraday's law of electromagnetic induction.

  1. All flowcharts need to be revised. Flowcharts have a beginning and an end, and all steps have a logical, interconnecting sequence. What it presents are just boxes of functional modules without realizing what makes it pass from one stage to another.

Our replies:

We thank the reviewer’s constructive advice. We have revised all flowcharts with beginning and an end; in addition, all steps become logical and interconnected.

Our revisions:

Please see flowcharts in Figs. 5a, 6a and 7a in the revised manuscript.

  1. The hardware description needs to be made, with clear evidence of the controller used, characteristics and processes carried out.

Our replies:

We thank the reviewer’s constructive comment. We have added the Table 2 for the hardware description, clear evidence of the controller used, characteristics and so on.

Our revisions:

Table 2 List of control and drive systems for multifunctional detection robot

Location

Name

Model

Parameter

Brand

Onboard control system

Power supply battery

6-EVF-32

12V 32Ah

Tian Neng

Battery Charger

Local

Suitable for 48V 32Ah batteries

Chao Wei

PLC

1769-L16ER-BB1B

Dual Ethernet w/DLR capability, 6 I/O Expansion via 1734 POINT I/O,24VDC

AB

I/O

1734 series

1*8-point switching input and output, 1*4-point analog input and output

AB

Servo Driver

DCPC-09050-E

DCPC-09050-E

DMK

Servo motor

130M-09525C5-EB

2KW ,48V,2500RPM,6NM

DMK

Gear reducer

PLX142-L3-120-S2-P2

120:1 planetary reducer

DMK

Wireless station

WLAN5100

IEEE 802.11 a/b/g/n, WLAN access point, supporting 2.4G and 5G, 300Mbps

Phoenix

Camera

GoPro

High definition 4K motion camera

GoPro

Wireless image transmission

Local

500~700M transmission

Freecast

Handheld operator

Power supply battery

Local

24V 8Ah

Local

Battery Charger

Local

Suitable for 24V 8Ah batteries

Local

Touch computer

PPC-3100S

10 inch touch screen, dual Ethernet card, Win 7

Adventach

Wireless station

WLAN1100

IEEE 802.11 a/b/g/n, WLAN access point, supporting 2.4G and 5G, 300Mbps

Phoenix

Wireless image transmission

Freecast CP3039

500~700M transmission

Freecast

  1. Inspection is limited to viewing an image from a GoPro or an analysis from an ultrasound system. Where is the automatic fault identification?

Our replies:

We thank the reviewer’s constructive comment. We agree with the reviewer that inspection is limited to viewing an image from a GoPro and ultrasound system. This current version of multifunctional detection robot for roller coaster track relies on manual assistance. We have stated this limitation clearly in conclusion.

Our revisions:

In conclusion, we added:

It should be noted that due to the lack of automatic fault identification, the current version of the detection robot relies on manual assistance.

  1. Avoid writing in the first person - "We"

Our replies:

We thank the reviewer’s constructive advice. The first person "We" has been revised in paper.

  1. The structure of the paper should be written at the end of the introduction;

Our replies:

We thank the reviewer’s constructive advice. 7. The structure of the paper has been added at the end of the introduction.

Our revisions:

In Introduction, we added:

In Section 2, the mechanical and the electrical control system of multifunctional roller coaster track detection robot with different sensors are established and tested. Finally, the conclusions are drawn in Section 3.

  1. Figure 1, identification of each subfigure with a), b), c) and d)

Our replies:

We thank the reviewer’s constructive advice. We have revised Figure 1 in the manuscript.

Round 2

Reviewer 2 Report

The questions asked in the first review have been clarified and the changes that were made is also aligned to the issues asked.

As suggestions for future work it's encouraged to work better in the component of inspection based on image.